# Autogenetic Gravity Center Placement

**DOI:** 10.3390/s25123786

**Published:** 2025-06-17

**Authors:** Timothy Sands

**Affiliations:** 1Department of Mechanical Engineering (SCPD), Stanford University, Stanford, CA 94305, USA; dr.timsands@alumni.stanford.edu; 2Department of Mechanical Engineering (CVN), Columbia University, New York, NY 10027, USA; 3Sibley School of Mechanical and Aerospace Engineering, Cornell University, Ithaca, NY 14853, USA

**Keywords:** space drone, experimental validation, center of gravity, auto-location, inertia identification, nonlinear Luenberger observers, adaptive, estimation, autonomy, control

## Abstract

**Highlights:**

Just this March, a promising, novel method to ascertain the location of the center of gravity of space systems was proposed, where mass inertia identification was also provided (both in real time). The promising technique was just validated in spaceflight tests.

**What are the main findings?:**

Real-time inertia identification proved very rapid (in fractions of a second).Autogenetic mass center location was also rapid, resolving in mere seconds.Relatively poor (decimal-degree-to-single-degree accuracy) state estimation was necessary.

**What is the implication of the main finding?:**

The validated novel method produces larger convergence than the modern comparative benchmark method.The modern comparative benchmark method was considerably slower.Dynamic space operations, like refueling, repair, grappling, and manipulation, are enhanced.

**Abstract:**

Operations by space drones mandate significant autonomy. This study experimentally evaluates key proposed applications of autonomy. Center of gravity auto-location is proposed using autonomous identification of mass properties, necessitating nonlinear state estimation. Nonlinear, coupled governing kinetics are strictly adopted as the control, and inversion provides closed-form estimates of mass properties. Seminally neglecting the diagonal inertia moments, the inertia cross-products are utilized to exactly find the mass center coordinates using the parallel axis theorem to parameterize the location coordinates. In December 2024, experiments were performed in space for hours, validating the approaches proposed. The findings indicate the longitudinal distribution was quite symmetric. Meanwhile, the lateral distribution was quite off-balance. Estimation convergence of the mass center coordinates was improved compared to the state-of-the-art comparative benchmark. In hundreds of days, the latter achieved millimeter convergence, while in minutes, the former achieved hundreds of millimeters convergence.

## 1. Introduction

Novel space maneuvers began in late 2024 by the X–37B reusable space drone [1], depicted in Figure 1, having already flown several innovative missions [2,3]. Meanwhile, the era of on-orbit refueling and repair with autonomous inspection is underway with free-flying space drones [4].

Such space drones necessitate autonomy on the vehicle, while other space drones are payloads, like the space robots depicted in Figure 2. Grappling robots are attached to the larger space drone vehicle; for example, the Dextre depicted in Figure 3 is attached to the refueling mission akin to Boeing’s MQ-25 Stingray aerial refueling drone [8].

While stochastic autonomy is presently burgeoning [15,16], closed-form analysis retains the benefits of implementation speed and ease of producing closed-form proofs of stability, convergence, and other key features. Autonomous space drones must autonomously determine mass properties on orbit, where, presumably, mass properties change drastically and rapidly.

### 1.1. How Conventional Methods Measure the Center of Mass Position and Inertia Parameters

Ubiquitously, development teams calculate inertia matrices during development and very often perform validating ground experiments, particularly revealing the diagonal matrix components, the mass moments of inertia. Discernment of off-diagonal products of inertia is relatively rarer.

### 1.2. Required Input Conditions

On-orbit inertia estimation was proposed using a least squares estimator weighted by rate sensor scale factors, and cost was computed using both angle and rate errors for use by MATLAB^®^’s *fmincon* optimization solver. The proposal was flown on MOST satellites in 2003, and, in July 2007, five attitude slew maneuvers were flown, validating the algorithm’s performance [17].

### 1.3. Specific Implementation Approaches

Arguably, principal components analysis applied to observational satellite imagery [18] leads thinking towards diagonal components. Computer-aided design tools were used [19,20,21] to estimate moments of inertia (the diagonal matrix terms) of two LAGEOS international laser ranging service satellites. Center of mass estimation was also performed [22]. Late studies addressed accelerometer anomalies by incorporating various types of disturbances [23]. State estimates (whose importance is highlighted in this present study) were similarly emphasized by the most recent studies [23]. The LAGEOS missions established a comparative benchmark as the current state of the art [24].

### 1.4. Existing Limitations

The performance of the state-of-the-art comparative benchmark may be considered the existing limitation. Center of gravity location estimation convergence is merely millimeter convergence over hundreds of days, exemplifying limitations in both speed and accuracy. Problem statements, research objectives, and key contributions are in Box 1.

Box 1Problem statements, research objectives, and key contributions.**Problem statements**: What are the time-varying mass and mass moments of space drones during operations that expend fuel (mass) and grapple with objects of unknown masses? Can these quantities be discerned using only angular displacement sensor data?**Research objectives**: (1) Reveal imbalanced mass exclusively utilizing rotational displacement data and (2) identify mass center offset using that imbalance and then articulate the coordinates of the mass center.**Key contributions**: Compared to a very recent comparative benchmark [25], the estimation of mass center coordinates is
improved from taking hundreds of days to achieve millimeter convergence to hundreds of millimeters convergence in mere minutes.

## 2. Materials and Methods

The X-37B orbital test vehicle (depicted in Figure 4) is a benchmark system that might operationally grapple targets to retrieve, replace, repair, or refuel. Such operational uses require indigenous, rapid learning of new system parameters like mass, mass moments and products of inertia, and the position of the center of gravity.

Another mission context for center of mass auto-location for grappling space robotic drones includes the U.S. Naval Academy (Figure 5), which provided a worthy benchmark launched into space in December 2018. Two 3D-printed manipulator arms are attached to the RSAT–P (USNA–19) cubesat on the NASA ELaNa XIX space mission.

### 2.1. Autonomous Control by Adopting Governing Kinetics as the Control

Equation (1) is the very well-known Euler’s moment equation governing rotational motion of masses. Ubiquitously, the equation is linearized, diagonalized, and simplified, permitting regression parameterization, where the motion states are declared the unknown vector of regression states. Less frequently, the mass moments and products of inertia are declared the unknown vector of regression states, permitting exact regression parameterization without linearization, diagonalization, or simplification, as shown in Equation (2). Where terms are defined in Table 1 (and similarly for Table 2 and Table A1).(1)$$\tau =\left[J\right]\left\{\dot{\omega }\right\}+\left\{\omega \right\}\times \left[J\right]\left\{\omega \right\}$$

Notice that Equation (2) necessitates knowledge of angular velocities and accelerations, either prescribed states to facilitate feedforward or knowledge of current states for feedback. Enhanced, nonlinear Luenberger observers (depicted in Appendix A) [28] are utilized for this purpose, where the observers accept inputs of angle-only and output torque estimates, angular acceleration estimates, and angular velocity estimates. The observer’s time-invariant gains may be tuned such that angle estimates equal input angles from sensors.(2)$$\left\{\begin{array}{c} u_{x}\\ u_{y}\\ u_{z} \end{array}\right\}\equiv \underset{{\left[\Phi \right]}_{d}}{{\underbrace{\left[\begin{array}{cccccc} {\dot{\omega }}_{x} & {\dot{\omega }}_{y}-\omega_{x}\omega_{z} & {\dot{\omega }}_{z}+\omega_{x}\omega_{y} & -\omega_{y}\omega_{z} & \omega_{y}^{2}-\omega_{z}^{2} & \omega_{z}\omega_{y}\\ \omega_{x}\omega_{z} & \omega_{y}\omega_{z}+{\dot{\omega }}_{x} & \omega_{z}^{2}-\omega_{x}^{2} & {\dot{\omega }}_{y} & {\dot{\omega }}_{z}-\omega_{x}\omega_{y} & -\omega_{x}\omega_{z}\\ -\omega_{x}\omega_{y} & \omega_{x}^{2}-\omega_{y}^{2} & {\dot{\omega }}_{x}-\omega_{y}\omega_{z} & \omega_{x}\omega_{y} & {\dot{\omega }}_{y}+{\omega_{x}\omega }_{z} & {\dot{\omega }}_{z} \end{array}\right]}}_{d}}\underset{\hat{\beta }}{\underbrace{\left\{\begin{array}{c} {\hat{J}}_{xx}\\ {\hat{J}}_{xy}\\ {\hat{J}}_{xz}\\ {\hat{J}}_{yy}\\ {\hat{J}}_{yz}\\ {\hat{J}}_{zz} \end{array}\right\}}}$$

### 2.2. Autonomous Inertia Identification Using Projection-Based Learning

Ubiquitously, nonlinear adaption [29] may be used to estimate inertia components. Inertia adaption has recently been combined with nonlinear dynamic inversion control for manipulators [30]. Instead, here, dynamic–optimal estimates (in a two-norm sense) are preferred for increased accuracy. Inverting the autonomous control Equation (2) provides solutions for inertia matrix components assembled into the variable $\hat{\beta }$ in accordance with Equation (3). Nominally, torque estimates from the Luenberger observer may be used in Equation (3) or, alternatively, so-called enhanced Luenberger observers may be used, where the known torque commands are utilized in a feedforward sense.(3)$$\hat{\beta }={\left({{\left[\Phi \right]}_{d}}^{T}{\left[\Phi \right]}_{d}\right)}^{-1}{\left[\Phi \right]}_{d}\hat{\tau }\;$$

Lastly, we consider center of gravity auto-location using the estimates provided by Equation (3).

### 2.3. Center of Gravity Auto-Location Parameterized by the Parallel Axis Theorem

By assuming the center of gravity is at some unknown location (not at the center of the body-fixed reference frame), the parallel axis theorem may be used to exactly locate the center of gravity without approximation. The analytic development is provided in Appendix B, leading to Equation (4).(4)$$\begin{array}{ccc} x_{cm}=\sqrt{\frac{J_{xy}J_{xz}}{mJ_{xy}}}, & y_{cm}=\sqrt{\frac{J_{xy}J_{yz}}{mJ_{xz}}}, & z_{cm}=\sqrt{\frac{J_{xz}J_{yz}}{mJ_{xy}}} \end{array}$$

Section 2 invoked Euler’s moment equation to exactly parameterize the control and then inverted the control and used nonlinear, optionally enhanced Luenberger observers [28] to estimate the variable necessary to exactly auto-locate the center of gravity. The only input needed is rotation angle sensor data.

### 2.4. Pseudocode Summarizing the Auto-Location Algorithm

Adopt the governing kinetics as the control.Formulate the controlled kinetics into exact regression parameterization.Invert the control expressed in regression form, isolating the inertia matrix components (both moments and cross-products).Use angular velocity and angular acceleration estimates from nonlinear Luenberger observers to solve for the inertia matrix components (both moments and cross-products).Expand the parallel axis theorem, solving three equations for three unknown position coordinates of the center of gravity parameterized in the off-diagonal inertia cross products alone.Use the off-diagonal inertia cross products alone for the auto-location of the center of gravity at every timestep.

## 3. Results

In December 2024, hours of on-orbit spaceflight experiments were performed, validating the analysis elaborated in Section 2 of this manuscript. The utilized satellite was asymmetric, and previous maneuvers indicated off-balance conditions, leading to the opinion that the center of gravity location was poorly understood.

### 3.1. Spaceflight Test Maneuvers

The angular rates and quaternions of slew maneuvers performed by the spacecraft are depicted in Figure 6. Figure 6a contains a three-dimensional plot whose axes correspond to body-fixed axes labeled as Euler axes (roll, pitch, and yaw). Figure 6b,c display quaternions comprising three imaginary quantities and one real quantity (rotation angle).

Meanwhile, the corresponding Euler angles are depicted in Figure 7 for maneuvers occurring over several hours in December 2024. All three angles are simultaneously displayed in Figure 7a, permitting immediate comparative situational awareness, while the roll, pitch, and yaw angles are separately plotted in Figure 7b, sharing only the display of time on the abscissa.

Angle state estimation errors (of the nonlinear, enhanced Luenberger observers) are depicted in Figure 8. Notice that the estimated accuracy is merely –1.2756, 1.3937, and −0.1174 degrees, respectively, for roll, pitch, and yaw (not very good!). The gains were rapidly tuned in the field to this first order using basic control gain tuning principles: (1) proportional gain was increased to produce convergence, (2) derivative gain was increased to reduce overshoot, and (3) integral gain was very cautiously increased to yield zero steady-state errors. Following data analysis, the reality emerged that impressive convergences were achieved despite poor observer performance, indicating that the algorithm is insensitive to estimation errors. The observer gains were not subsequently retuned, seeking improved performance.

Section 3.1. reported the experimental maneuver data from spaceflight experiments flown in December 2024. Meanwhile, the data analysis results are produced subsequently. Section 3.1 displays the results of the exact calculations of estimates for inertia moments and products. Section 3.2 uses those estimates to exactly calculate the location of the center of gravity.

### 3.2. Inertia Identification

Figure 9 displays the estimated moments and products of inertia using Equation (3). Despite the unimpressive attitude estimation accuracy of the Luenberger observers, all the inertia estimates converge rapidly to new values consistently.

Notice that the initial inertia estimates initiate the plots in Figure 9. The immediate diverge from initial values upon maneuvering indicates the initial estimates were poor. The rapid convergence to new values indicates the effectiveness of the proposed projection-based learning method. The inertia estimates from Figure 9 may be used in Equation (4) to produce exact calculations for locating the center of gravity.

**Figure 9 sensors-25-03786-f009:**
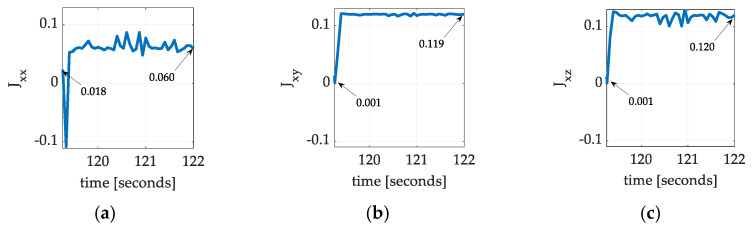
Projection regression-based learning inertia moments and products for spaceflight experiments flown in December 2024 corresponding to Table 3. Time varying mass moments and products for a single-axis slew maneuver with coupled motion in roll and pitch. (**a**) $J_{xx}$; (**b**) $J_{xy}$; (**c**) $J_{xz}$; (**d**) $J_{yy}$; (**e**) $J_{yz}$; and (**f**) $J_{zz}$.

**Table 3 sensors-25-03786-t003:** Summary of empirical inertia learning corresponding to Figure 9.

Inertia Component	Initial Guess	Converged Value	Percent Change
$J_{xx}$	0.018	0.060	233%
$J_{xy}$	0.001	0.119	11,800%
$J_{xz}$	0.001	0.120	11,900%
$J_{yy}$	0.026	0.060	131%
$J_{yz}$	0.001	0.119	11,800%
$J_{zz}$	0.006	0.058	867%

Raw data corresponding to Figure 9.

### 3.3. Center of Gravity Auto-Location

Using the estimated inertia cross-products alone (not the moments of inertia), Equation (4) produces exact calculations of the coordinates of the center of gravity. Figure 10, Figure 11 and Figure 12 display the results of the auto-location achieved experimentally with corresponding values in Table 4, Table 5 and Table 6 summarized in Box 2. The results indicate the mass is very well balanced about the spacecraft $\hat{x}$-axis but very imbalanced about the $\hat{y}$-axis and $\hat{z}$-axis.

Many test maneuvers were performed. Three of them, in disparate axes, are presented (in Figure 10, Figure 11 and Figure 12), where the maneuvers all converge to very similar estimated positions of the center of gravity. All three experiments indicate that the mass was initially quite symmetrically distributed about the $\hat{x}$-axis alone. Meanwhile, the mass distribution about the $\hat{y}$-axis and $\hat{z}$-axis was initially quite imbalanced. Luenberger observer estimates (not terribly well tuned) were used to estimate the requisite cross-products of inertia, producing convergent estimates of the location of the center of gravity.

Box 2Summary of results.**Key results**: Notice that in Figure 9, Figure 10 and Figure 12, despite disparate initial location guesses, both the experiments rapidly converge to identical locations (naturally producing disparate percent changes).

## 4. Discussion

The algorithm proposed necessitates only angle information from sensors to formulate all necessary quantities to auto-locate the space drone’s center of gravity. Intermediate calculations (nonlinear state estimation) necessitate user-selected calculation of angular velocities and acceleration. In this instance, nonlinear state estimation was selected using Luenberger observers, so the user-selected values in this instance included three observer gains, and converged values and percentage change are presented in Table 7 in Section 4. Discussion.

The algorithm uses exact, non-approximated relationships in all places except nonlinear state observation.

The governing kinetics were adopted as the (exact) control without typical approximations, like model-reference control, and classical feedback controls (e.g., proportional, integral, derivative), etc. The controlled kinetics are exactly parameterized in regression form and solved for time-varying, unknown inertia matrix components, and the regression is inverted to reveal the inertia matrix components in closed form. Inexact nonlinear motion observers were used in the inverted regression to produce inertia estimates. Lastly, the parallel axis theorem was used to produce three equations and three unknown coordinates of the center of gravity. The three equations were solved simultaneously to produce close-form exact equations for the auto-location of the center of gravity, where the off-diagonal inertia cross-products alone (without diagonal inertia moments) indicate the location by embodying mass imbalance.

The comparative benchmark most recently flown in space (in 2018) achieved millimeter convergence over hundreds of days (reported in the literature between 2019 and 2024). Meanwhile, the proposed algorithm was flown in space in December 2024, achieving tens of centimeters convergence in mere seconds. The proposed algorithm is closed-form, requiring no approximation or formulation and solution of an optimization problem. The main weakness remains the utilization of nonlinear state observers to estimate requisite motion states in the absence of dedicated sensors. Following the spaceflights flown by drones reported in this manuscript, the next logical recommended step is the application to grappling robotic space drones with rapidly changing mass properties.

The proposed method’s parameter identification efficiency requires comparison with conventional approaches. Additionally, the reliability of its converged values needs validation against benchmark standards or through simulation experiments to enable meaningful performance evaluation. Furthermore, differences between the initial and converged values of inertia/mass parameters may affect the final estimates, particularly with respect to the initial value. The former was accomplished in the prequel work [31,32,33], emphasizing analysis and simulation, but Monte Carlo analysis should include impacts of the initial condition as well as the performance sensitivity in response to varied sensor noise, especially since such noise is artificially induced in the prequel to maintain persistent excitation of estimation. The next stage of research for any such seemingly promising developments includes validation in repeated spaceflights, as well as generalization by application to disparate space systems. Both are included in the sequel research efforts, which will also include the recommended future research.

### Recommended Future Research

Improved nonlinear state estimation (analytic study).Performance in response to grappling unknown objects.Investigate converged values’ reliability using Monte Carlo analysis and simulations.A completely new study of highly flexible space robotics [31,34] should investigate if the developments presented here can be generalized to non-rigid body cases.

## Figures and Tables

**Figure 1 sensors-25-03786-f001:**
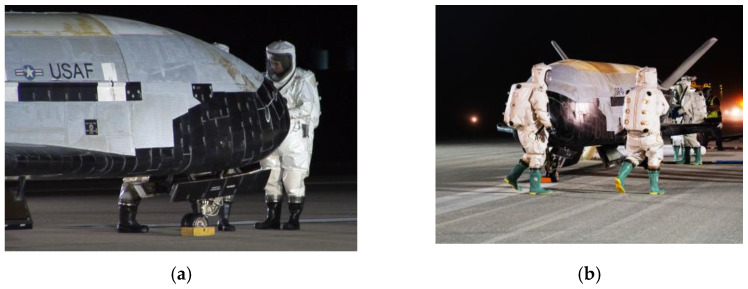
X-37B orbital test vehicle (**a**) lands at Vandenberg AFB on 3 December 2010 at 1:16 a.m. [5]. (**b**) Orbital test vehicle completes sixth successful mission [6]. Image credits: U.S. Space Force, used in compliance with published image use policy [7].

**Figure 2 sensors-25-03786-f002:**
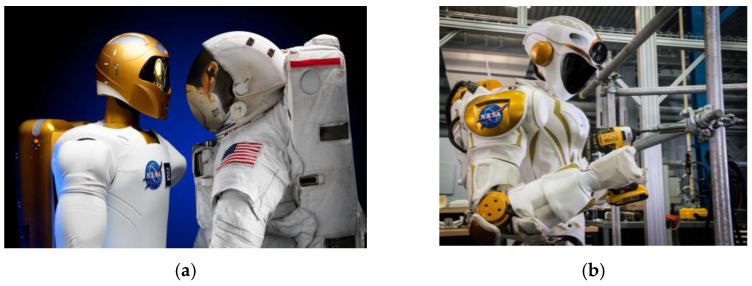
This is a figure. (**a**) Robonaut2 image credit: NASA [9]; (**b**) Valkyrie. Image credit: NASA [10]. Images used in accordance with image use policy [11].

**Figure 3 sensors-25-03786-f003:**
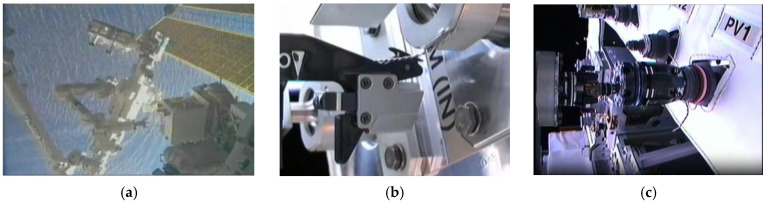
This is a figure. (**a**) Dextre approaches robotic refueling mission on 8 March 2011 with a wire cutter tool. Image credit: NASA [12]. (**b**) Robotic refueling mission’s Dextre drone successfully snipping two twisted, very thin wires with only a few millimeters clearance. Image credit: NASA [13]. (**c**) Robotic refueling mission separating from refueling receptacle with discharged propellant visibly floating away. Image credit: NASA [14]. Images used in accordance with the image use policy [11].

**Figure 4 sensors-25-03786-f004:**
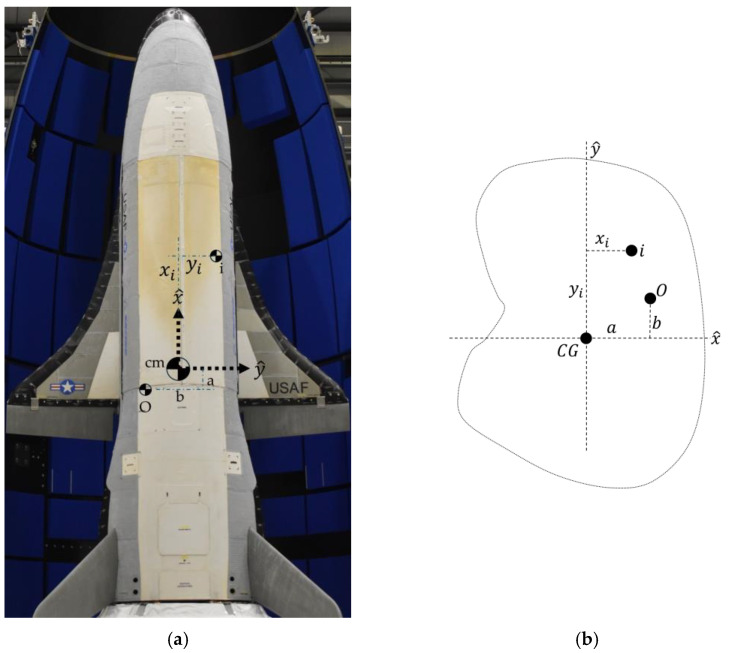
Spacecraft gravity center nomenclature: (**a**) sample spacecraft with multiple possible locations (small symbols) and large-symbolled actual location. (**b**) Schematic of center of mass offset from spacecraft body axes centered at the origin, O, corresponding to the parallel axis theorem elaborated in Appendix B. Image credit: Space Force [26], used in compliance with published image use policy [7].

**Figure 5 sensors-25-03786-f005:**
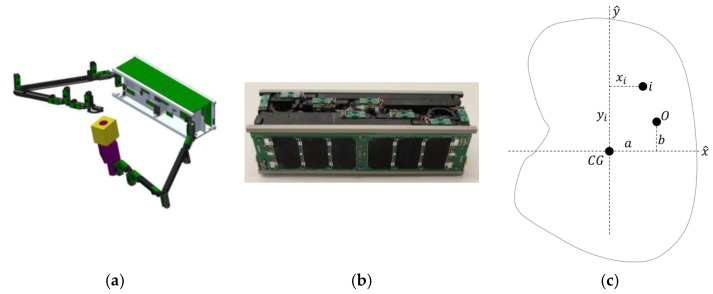
U.S. Naval cubesat grappling space robot (image credits: U.S. Navy), launched in 2018 [27]. Used in accordance with the published image use policy [7]. (**a**) Computer-aided drawing depicting two robotic arms with cameras on each end effector and at the cube satellite’s mid-body. (**b**) Actual hardware with robot arms stowed. (**c**) Schematic of the center of mass offset from the spacecraft body axes centered at the origin, O, corresponding to the parallel axis theorem elaborated in Appendix B.

**Figure 6 sensors-25-03786-f006:**
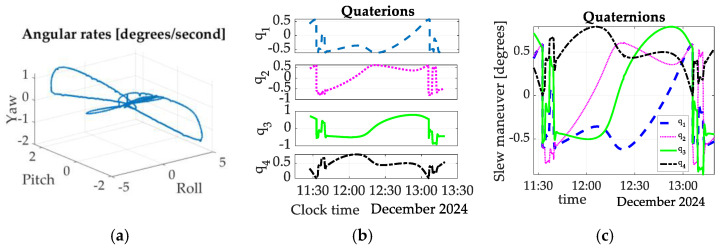
Spaceflight test maneuver. (**a**) Angular rates, quaternions, and Euler angles in degrees versus time in seconds for about two hours of spaceflight experiments flown in December 2024. (**b**) quaternions versus time; (**c**) quaternions versus time.

**Figure 7 sensors-25-03786-f007:**
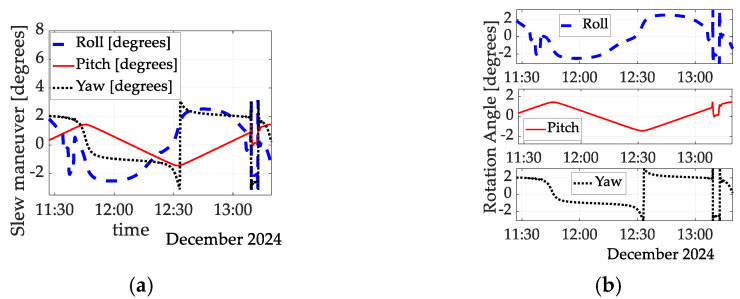
Euler angles in degrees versus time in seconds for spaceflight experiments flown in December 2024: (**a**) Euler angles versus time; (**b**) three separate plots for roll, pitch, and yaw angles.

**Figure 8 sensors-25-03786-f008:**
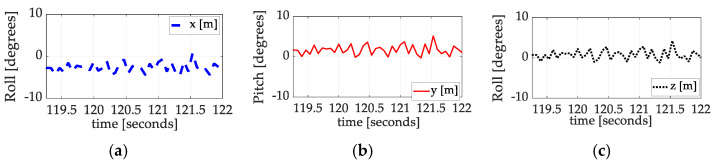
Luenberger state estimation errors for spaceflight experiments flown in December 2024: (**a**) roll
angle; (**b**) pitch angle; (**c**) z angle. Although the observer gains were coarsely tuned to unity-gain (the estimates of angled equal to the angles provided by the sensors), the observer estimation accuracy limit is displayed. These inaccuracies manifest as inaccurate inertia estimates, which, in turn, provide the limit upon accuracy of the auto-location of the center of mass. Future research should focus on nonlinear observer improvements.

**Figure 10 sensors-25-03786-f010:**
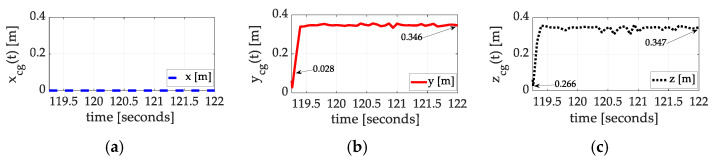
Exact calculations for the location of the center of mass for space experiments, set #2, flown on 20 December 2024. (**a**) x coordinate; (**b**) y coordinate converges from 0.028 cm to 34.60 cm; (**c**) z coordinate converges from 0.266 cm to 34.70 cm. This experimental maneuver was performed much later in the maneuver sequence than the maneuvers displayed in Figure 11.

**Figure 11 sensors-25-03786-f011:**
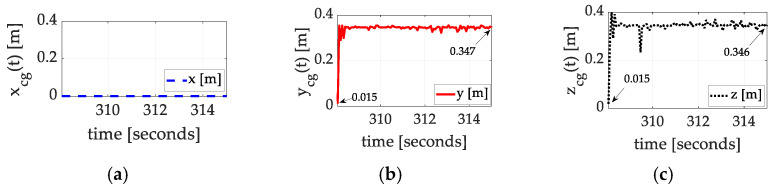
Exact calculations for the location of the center of mass for space experiments, set #2, flown on 20 December 2024. (**a**) x coordinate; (**b**) y coordinate converges from 0.015 cm to 34.70 cm; (**c**) z coordinate converges from 0.015 cm to 34.60 cm.

**Figure 12 sensors-25-03786-f012:**
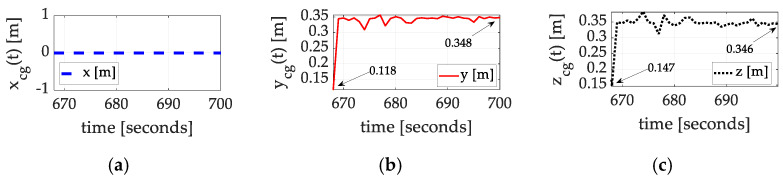
Exact calculations for the location of the center of mass for space experiments, set #3, flown on 20 December 2024. (**a**) x coordinate; (**b**) y coordinate converges from 0.0118 cm to 34.80 cm; (**c**) z coordinate converges from 0.147 cm to 34.60 cm. Notice that the descriptors have changes where a centimeter is $1\times 10^{-2}$. This change in presentation permits easy comparison to the comparative benchmarks [24,25] of this study.

**Table 1 sensors-25-03786-t001:** Table of proximal variables and nomenclature ^1^.

Variable/Acronym	Definition	Variable/Acronym	Definition
$\left[J\right]=J$	Inertia matrix or tensor	$J_{xx}$	Moment of inertia about $\hat{x}$
$\left\{\dot{\omega }\right\}=\dot{\omega }$	Angular acceleration	$J_{xy}=J_{yx}$	$\hat{x}\hat{y}$ inertia product
$\left\{\omega \right\}=\omega$	Angular velocity	$J_{xz}=J_{zx}$	$\hat{x}\hat{z}$ inertia product
${\dot{\omega }}_{x}$	Angular acceleration in the $\hat{x}$-direction	$J_{yy}$	Moment of inertia about $\hat{y}$
${\dot{\omega }}_{y}$	Angular acceleration in the $\hat{y}$-direction	$J_{yz}=J_{zy}$	$\hat{y}\hat{z}$ inertia product
${\dot{\omega }}_{z}$	Angular acceleration in the $\hat{z}$-direction	$J_{zz}$	Moment of inertia about $\hat{z}$
$\left[\Phi ?\right]=\Phi ?$	Regression matrix of knowns	$\left\{\beta ?\right\}=\beta$	Unknown, predicted variables

^1^ Such tables are offered throughout this manuscript to aid readability.

**Table 2 sensors-25-03786-t002:** Table of proximal variables and nomenclature ^1^.

Variable/Acronym	Definition	Variable/Acronym	Definition
${\left[\omega_{x}\right]}_{d}$	Desired angular velocity about $\hat{x}$	${\left[{\dot{\omega }}_{x}\right]}_{d}$	Desired angular acceleration about $\hat{x}$
${\left[\omega_{y}\right]}_{d}$	Desired angular velocity about $\hat{y}$	${\left[{\dot{\omega }}_{y}\right]}_{d}$	Desired angular acceleration about $\hat{y}$
${\left[\omega_{z}\right]}_{d}$	Desired angular velocity about $\hat{z}$	${\left[{\dot{\omega }}_{z}\right]}_{d}$	Desired angular acceleration about $\hat{z}$
$\left[\Phi \right]=\Phi$	Regression matrix of sensor data	${\left[\Phi \right]}_{d}=\Phi_{d}$	Regression matrix of desired states
β	Unknown, predicted variables	$\hat{\beta }$	Estimated variables
τ	Total control signal	$\tau_{ff}$	Feedforward control signal
${\hat{J}}_{xx}$	Estimated Moment of inertia about $\hat{x}$	${\hat{J}}_{xy}={\hat{J}}_{yx}$	Estimated $\hat{x}\hat{y}$ inertia product
${\hat{J}}_{yy}$	Estimated Moment of inertia about $\hat{y}$	${\hat{J}}_{xz}={\hat{J}}_{zx}$	Estimated $\hat{x}\hat{z}$ inertia product
${\hat{J}}_{zz}$	Estimated Moment of inertia about $\hat{z}$	${\hat{J}}_{yz}={\hat{J}}_{zy}$	Estimated $\hat{y}\hat{z}$ inertia product

^1^ Such tables are offered throughout this manuscript to aid readability.

**Table 4 sensors-25-03786-t004:** Summary of empirical inertia learning corresponding to Figure 10.

Inertia Component	Initial Guess	Converged Value	Percent Change
$x_{cg}$	0	0	0
$y_{cg}$	0.028	0.346	1136%
$z_{cg}$	0.266	0.347	30%

Raw data corresponding to Figure 10.

**Table 5 sensors-25-03786-t005:** Summary of empirical inertia learning corresponding to Figure 11.

Inertia Component	Initial Guess	Converged Value	Percent Change
$x_{cg}$	0	0	0
$y_{cg}$	0.015	0.347	2213%
$z_{cg}$	0.015	0.346	2207%

Raw data corresponding to Figure 11.

**Table 6 sensors-25-03786-t006:** Summary of empirical inertia learning corresponding to Figure 12.

Inertia Component	Initial Guess	Converged Value	Percent Change
$x_{cg}$	0	0	0
$y_{cg}$	0.118	0.348	195%
$z_{cg}$	0.147	0.346	135%

**Table 7 sensors-25-03786-t007:** Percent performance improvements.

Parameter	Convergence Percentage	Converged Value	Convergence Time [Seconds]
$J_{xx}$	233%	0.060	3
$J_{xy}$	11,800%	0.119	
$J_{xz}$	11,900%	0.120	
$J_{yy}$	131%	0.060	
$J_{yz}$	11,800%	0.119	
$J_{zz}$	867%	0.058	
Space test #1 $x_{cg}$	0	0	
Space test #1 $y_{cg}$	1136%	0.347	
Space test #1 $z_{cg}$	30%	0.346	
Space test #2 $x_{cg}$	0	0	7
Space test #2 $y_{cg}$	2213%	0.347	
Space test #2 $z_{cg}$	2207%	0.346	
Space test #3 $x_{cg}$	0%	0	32
Space test #3 $y_{cg}$	2213%	0.347	
Space test #3 $z_{cg}$	2207%	0.346	

## Data Availability

The dataset is available on request from the corresponding author.

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
