# Peer review of "Autogenetic Gravity Center Placement"

_sensors, 2025, doi:10.3390/s25123786_

Round 1

Reviewer 1 Report

Comments and Suggestions for Authors

This paper proposes an on-orbit rapid identification method for time-varying mass and inertia properties of spacecraft, including both inertia tensor and center-of-mass position. The method relies solely on attitude angle sensor data and leverages Euler's moment equations to achieve inertia parameter identification. The technical feasibility of this approach has been demonstrated through on-orbit validation. However, the following issues persist in this paper:

  1. Reference [5] appears uncited in the paper.
  2. The figure title (c) in Figure 6 seems incorrect.
  3. Equations (B21) and (B22) in the Appendix are duplicates.
  4. The Introduction should provide detailed explanations regarding: (1) how conventional methods measure center-of-mass position and inertia parameters on orbit, (2) their required input conditions, (3) specific implementation approaches, and (4) existing limitations."
  5. While the paper analyzes the differences between initial and converged values of inertia/mass parameters, it omits examination of whether initial values affect the final estimates. We recommend adding analysis on how different initial estimates might impact the converged parameter values.
  6. The proposed method's parameter identification efficiency requires comparison with conventional approaches. Additionally, the reliability of its converged values needs validation against benchmark standards or through simulation experiments to enable meaningful performance evaluation.

Author Response

  • Reference [5] appears uncited in the paper.

RESPONSE: Thanks for the great catch.  Citation of [4] was erroneously placed where [5] belonged.  The mistake has been corrected in the revision.

  • The figure title (c) in Figure 6 seems incorrect.

RESPONSE: Indeed.  Great catch.  Thanks again.  The caption has been revised.

  • Equations (B21) and (B22) in the Appendix are duplicates.

RESPONSE: Thanks for noticing the similarity. The first (equation (21)) culminates a step-by-step development process, while equation (22) merely introduces the labeling of new vector and matrix variables (annotated by under–bracket accents) carried forward.

  • The Introduction should provide detailed explanations regarding: (1) how conventional methods measure center-of-mass position and inertia parameters on orbit, (2) their required input conditions, (3) specific implementation approaches, and (4) existing limitations."

RESPONSE: Thanks for the good recommendation.  The request is ceded, and the revised manuscript highlights these details.

  • While the paper analyzes the differences between initial and converged values of inertia/mass parameters, it omits examination of whether initial values affect the final estimates. We recommend adding analysis on how different initial estimates might impact the converged parameter values.

RESPONSE: Great idea. The current elaboration of this facet in section 3 and highlighted in Box 2 has now been augmented in the revision. Further elaborated specification has been added to section 4, especially since initial values were not truly iterated (e.g. Monte Carlo) in the prequel analysis and simulations.

  • The proposed method's parameter identification efficiency requires comparison with conventional approaches. Additionally, the reliability of its converged values needs validation against benchmark standards or through simulation experiments to enable meaningful performance evaluation.

RESPONSE: Great suggestion. The revised manuscript addresses these issues in section 4.

Reviewer 2 Report

Comments and Suggestions for Authors

Suggested revisions for manuscript ID: sensors-3678973

      The manuscript presents an algorithm that can real - time calculate the center of mass and inertia matrix only using the data from attitude angle sensors. This solves the problem of changes in mass properties caused by fuel consumption or grabbing objects in space operations. Secondly, the proposed algorithm has a significant improvement in efficiency. Compared with the existing methods that need hundreds of days to achieve millimeter - level accuracy, the new method can achieve centimeter - level convergence in just a few minutes. In addition, by strictly using non - linear control equations and the parallel axis theorem, the manuscript realizes a closed - form solution without approximation. Especially, the method of using non - diagonal inertia products to accurately locate the center of gravity is very innovative. The manuscript as a whole has high research value and academic level. Although the manuscript has achieved great results, there are still some problems that need further modification or explanation, as follows:

  1. The description of some data is not unified. For example, in the description of Figure 10, the markers all use "m" as the unit, but when analyzing Figure 10, the unit is converted to "cm" for description. Please explain the reason for this description or unify the description method. Similarly, please correct similar problems in the full text.
  2. The gain tuning logic of the non - linear Luenberger observer is not explained. Only "coarsely tuned" is mentioned (Page 7), and the influence of estimation error on the results is not analyzed. Please explain or consider adding the analysis.
  3. In the abstract, there is a description of "only requiring attitude angle data", but in reality, it relies on the estimated values of angular velocity/angular acceleration (from the observer). It is necessary to clearly state that the input is the original sensor data and the intermediate state is the estimated value.
  4. For the influence of quantization error, Monte Carlo analysis can be considered to evaluate the sensitivity of sensor noise to the positioning accuracy of the center of gravity.
  5. It is recommended to add "verification of the algorithm on non - rigid spacecraft (such as flexible solar wings)" in "Recommended future research".

Author Response

  • The description of some data is not unified. For example, in the description of Figure 10, the markers all use "m" as the unit, but when analyzing Figure 10, the unit is converted to "cm" for description. Please explain the reason for this description or unify the description method. Similarly, please correct similar problems in the full text.

RESPONSE: Great catch.  The rationale for the modified presentation was not highlighted, and is now described in the caption of figure 12 in the revision.

  • The gain tuning logic of the non - linear Luenberger observer is not explained. Only "coarsely tuned" is mentioned (Page 7), and the influence of estimation error on the results is not analyzed. Please explain or consider adding the analysis.

RESPONSE: Thanks for the great suggestion.  Elaboration has been added to section 3.1.

  • In the abstract, there is a description of "only requiring attitude angle data", but in reality, it relies on the estimated values of angular velocity/angular acceleration (from the observer). It is necessary to clearly state that the input is the original sensor data and the intermediate state is the estimated value.

RESPONSE: Indeed.  Thanks for the great suggestion.  This request is accommodated with revised text in section 4 elaborating this important distinction.  Thanks again.

  •  For the influence of quantization error, Monte Carlo analysis can be considered to evaluate the sensitivity of sensor noise to the positioning accuracy of the center of gravity.

RESPONSE: Great suggestion.  The revised manuscript addresses these issues in section 4.

  • It is recommended to add "verification of the algorithm on non - rigid spacecraft (such as flexible solar wings)" in "Recommended future research".

RESPONSE: Great idea, thanks.  Such has been added to section 4.1 including the flexible robotics prequels permitting the readership to see the convergence of both lines of research.

Round 2

Reviewer 1 Report

Comments and Suggestions for Authors

The author has responded to and partially corrected the issues I raised earlier, while listing some of them as future research topics, which is acceptable. However, there appears to be an error in the title of Figure 1 . I recommend the author double-check it.

Author Response

Thanks very kindly for the Reviewer's diligence.  The accidental omission included the citation of the image use policy (a major omission). The remedy is gratefully included in the re-revision.  

Reviewer 2 Report

Comments and Suggestions for Authors

All concerns have basically been resolved, and there's nothing more to comment on.

Author Response

Thanks for a very worthwhile review.  The manuscript is certainly improved thanks to the Reviewer's efforts.